# A Pre-Exposure to Male-Specific Compound *γ*-Hexalactone Reduces Oviposition in *Bactrocera oleae* (Rossi) (Diptera: Tephritidae) Under Laboratory Conditions

**DOI:** 10.3390/insects16020147

**Published:** 2025-02-01

**Authors:** Sergio López, Clàudia Corbella-Martorell, Elisa Tarantino, Carmen Quero

**Affiliations:** Department of Biological Chemistry, Institute for Advanced Chemistry of Catalonia (IQAC-CSIC), Jordi Girona 18-26, 08034 Barcelona, Spain; claudia.martorell@iqac.csic.es (C.C.-M.); eltiuv@cid.csic.es (E.T.); carme.quero@iqac.csic.es (C.Q.)

**Keywords:** *Bactrocera oleae*, *γ*-hexalactone, oviposition activity index, pre-exposure

## Abstract

Laboratory trials have been undertaken to determine the effect of the pre-exposure to *γ*-hexalactone, a specific compound released by virgin males of *Bactrocera oleae* (Rossi) (Diptera: Tephritidae), on the oviposition activity of the species. After pre-exposing virgin females to 1.0 mg of *γ*-hexalactone for 24 h, we observed that the number of eggs laid on an artificial substrate after mating was significantly reduced, whereas no significant differences were observed with a previous experience to 0.5 mg of compound. These results are of particular interest for delving into the possibility of using intraspecific semiochemicals as oviposition disrupting tools in the olive fruit fly, although more research is still needed to gain a deeper knowledge about the effect of a previous experience to *γ*-hexalactone on the behavior of the species.

## 1. Introduction

The olive fruit fly *Bactrocera oleae* (Rossi) (Diptera: Tephritidae) is regarded as the most threatening pest insect for olive trees (*Olea europaea* L.) worldwide, especially in the Mediterranean region [1]. It is a monophagous and multivoltine species, in which a female can lay 200–400 eggs during its life. After laying a single egg per olive [2], the larva feeds on the mesocarp and passes through three larval instars before pupation. Egg laying and larval feeding lead to a depreciation on both table fruit and olive oil production and quality, due to premature fruit drop, larval consumption of fruit pulp and oil quality deterioration by severe changes in the chemical composition of the fruit [3,4,5,6]. This damage is translated into significant economic losses [7], and consequently high investments are yearly destined to control olive fruit fly populations. To give an example, the annual costs invested by eight Mediterranean countries (i.e., Spain, France, Italy, Portugal, Greece, Cyprus, Slovenia, and Malta) in pesticide applications are estimated in €5 billion [8].

In contrast to other tephritid species, in which males are the responsible of releasing the sex pheromone, olive fruit fly females carries all the chemical burden involved in sexual communication [9,10]. However, male-mediated chemical attraction upon females has been reported, suggesting thus the existence of male-specific olfactory stimuli [11,12,13,14]. First evidence of male-mediated chemical attraction on females was provided by De Marzo and coworkers, who observed that rectal gland secretions from sexually mature males attracted females in olfactometric trials [14]. Further studies revealed that solvent extractions from sexually mature male bodies resulted to be attractive for females during the last two hours of the photophase [11]. Later, the unsaturated hydrocarbon (*Z*)-9-tricosene (trivially known as muscalure) was identified from rectal glands of males, and it attracted females at a close range [12]. Recently, López and coworkers identified two lactones, namely *γ*-hexalactone and *δ*-hexalactone, from the volatile bouquet released from virgin males of different ages (i.e., from 1 to 23 days old), with the former resulting attractive for both sexes in double-choice olfactometer trials [13]. In addition to this positive chemotaxis, the number of field catches of both sexes to the food lure ammonium bicarbonate is increased in presence of *γ*-hexalactone, and these binary mixture also results more attractive than the combination of the ammonium salt and 1,7-dioxaspiro[5.5]undecane (hereafter referred as olean), the major sex pheromone component of the species [13]. These findings on the chemical intraspecific communication of the species have paved the way for developing novel strategies for monitoring and/or mass trapping *B. oleae* populations, at the same time that arises questions regarding the biological role of *γ*-hexalactone on the species. Given that a previous experience to either sex pheromones or plant volatiles is known to affect the behavioral and physiological response of insects in different contexts [15,16,17,18,19,20], we hypothesized that *γ*-hexalactone may induce a change in the reproductive behavior of *B. oleae* females after being pre-exposed to the compound. For that purpose, we designed a specific laboratory assay to test whether the oviposition rate on an artificial substrate is altered after pre-exposing virgin females to *γ*-hexalactone for a period of 24 h. If adversely affected, the disruption of oviposition through a species-related olfactory stimulus may become a promising key point for the development of novel control approaches against the olive fruit fly.

## 2. Materials and Methods

### 2.1. Insect Rearing

All the flies were obtained from a laboratory colony at the installations of the Institute for Advanced Chemistry of Catalonia (Barcelona, Spain). This permanent colony has been maintained since 2016, and it was initially set up from a parental generation obtained from the Joint FAO/IAEA Centre of Nuclear Techniques in Food and Agriculture (Vienna, Austria). Rearing and maintenance procedures in our installations followed previously described methodologies. Adults were kept in cubic Bugdorm© cages (30 × 30 × 30 cm, Entompraxis, Barcelona, Spain), in which one of the walls had been replaced by a mesh covered with a mix of paraffins and bee wax to act as oviposition substrate [21]. Laid eggs were daily collected by rinsing this wall with water, and they were subsequently left for 48 h on a fine mesh upon a sponge wet with a 0.3% propionic acid solution [22]. Afterwards, eggs were washed with water and placed in Petri dishes (15 cm diameter) containing the larval diet [23].

To obtain virgin individuals of both sexes (7–14 days old) for being further tested, the presence of pupae was daily checked, and upon emergence, adults were sorted by sex, and those of the same sex were pooled in Bugdorm© cages. Adults were fed on a mixture of sugar, yeast hydrolysate, and egg yolk (75:19:6) [23], and water was provided *ad libitum*, by wetting a sponge strip. Both food and water were replaced every two days. All the developmental stages were kept at 24 ± 1 °C, 55 ± 5% RH, and a L:D photoperiod of 16:8.

### 2.2. Pre-Exposure to γ-Hexalactone

For pre-exposing virgin olive fruit fly females to two different release rates, *viz*. 0.5 and 1.0 mg/24 h, of racemic *γ*-hexalactone (98%, Alfa Aesar, Heysham, United Kingdom), a 2 mL-polyethylene capsule filled with 1 mL of the compound was used. As a single capsule provided an estimated release rate of 0.5 mg/24 h, two capsules were used to obtain a release rate of 1.0 mg/24 h. This release rate of *γ*-hexalactone had been previously determined by weighting a dispenser (n = 3 replicates) daily during two weeks under laboratory conditions, and subsequently calculating the negative slope from the regression analysis of obtained weights.

In each sample, four females were introduced in a disposable plastic glass (400 mL), with the top of the glass covered with a paper towel, and the capsule containing *γ*-hexalactone was hung from the napkin using a nickel paper clip. Water and sugar were provided *ad libitum* from a moistened cotton ball placed in the bottom of the glass. Control flies were kept under the same conditions in absence of the dispenser releasing *γ*-hexalactone. With regard to males, virgin individuals (7–14 days old) were kept in a cubic Bugdorm© cage and fed on water and sugar prior to pairing them with females in further oviposition assays. Both sexes were kept under the same conditions (24 ± 1 °C, 55 ± 5% RH, 16:8 L:D photoperiod).

### 2.3. Oviposition Assays

Pre-exposed and control females were separately transferred to a customized oviposition arena after 24 h. Briefly, this oviposition arena consists of two Petri dish bases (15 cm diameter), with one of those put upside down upon the other, and laterally sealed with sticky tape. A wooden stretcher holding the same paraffin-covered mesh (9 cm diameter) was used as oviposition substrate. Flies were allowed to feed on a cotton ball moistened with water and sugar during the duration of the assay. The number of laid eggs were counted after 24 h under the stereomicroscope. A total of six replicates for each release rate of *γ*-hexalactone and 12 control samples (one per each *γ*-hexalactone replicate) were run under 24 ± 1 °C and 55 ± 5% of relative humidity.

### 2.4. Statistical Analysis

Due to the relative low sample number (n = 6 per release rate of *γ*-hexalactone) and to avoid any violation of the assumptions of normality and homoscedasticity, the mean number of eggs per female was analyzed with the non-parametric Kruskal-Wallis test followed by the Mann-Whitney *U* test after Bonferroni correction for pairwise comparisons.

The oviposition activity index (OAI), proposed by Kramer and Mulla [24], was also calculated. This index is given by the formula OAI = NT-NC/NT + NC, where NT and NC refer to the number of eggs found in the treatment and control, respectively. The value of this index ranges from 1.0 to −1.0, and according to Kramer and Mulla, values lower than −0.3 indicate that the treatment is reducing the oviposition, while those OAI higher than 0.3 suggest than oviposition is induced by the compound tested [24].

## 3. Results

A 24-h pre-exposure to *γ*-hexalactone significantly affected the oviposition rate of females (χ^2^ = 11.193, d.f. = 2, *p* = 0.004). A reduction in the number of eggs per female was observed after a previous exposure to 0.5 mg of *γ*-hexalactone (control: 22.4 ± 10.9; γ-hexalactone: 14.2 ± 6.3), albeit no significant differences were detected (*U* = 16.000 *Z* = −1.876, *p* = 0.067) (Figure 1). Conversely, a significant reduction occurred in those females pre-exposed to 1 mg of compound (control: 22.4 ± 10.9; *γ*-hexalactone: 6.8 ± 6.1) (*U* = 4.000, *Z* = −3.002, *p* = 0.001) (Figure 1). No significant differences were observed between the number of eggs of females pre-exposed to either amount of *γ*-hexalactone (*U* = 6.000, *Z* = −1.928, *p* = 0.065) (Figure 1). The OAI values from those females pre-exposed to 0.5 mg of racemic *γ*-hexalactone ranged from −0.13 to −0.50, with a mean average of −0.24 ± 0.17 (Figure 2), whereas a pre-exposure to 1.0 mg of compound yielded a mean OAI of −0.56 ± 0.22 (from −0.25 to −0.76) (Figure 2).

## 4. Discussion

In insects, a previous experience to an olfactory signal, either sex pheromones or plant volatiles, has widely been demonstrated to modify their behavior and activity [15]. Most of these studies have targeted lepidopteran species belonging to different families, in which identifying whether a previous olfactory signal may lead to sensitization or habituation/desensitization is key for a better understanding of the basis of mating disruption [25]. In this sense, several research works report that the responsiveness to either sex to the sex pheromone [16,17,18], courtship and mating behaviors [26,27] are affected following a pre-exposure to sex pheromone. This effect widely varies among species, and can be observed at both olfactory [16,28,29,30] and behavioral level [17,27,28,31].

To the best of our knowledge, few literature records have focused on how a previous experience to a pheromone influences the oviposition behavior of a species [32,33]. A research conducted on *Zygaena filipendulae* (L.) (Lepidoptera: Zygaenidae) showed that the number of eggs laid by those females pre-exposed to the sex pheromone did not differ to that of naïve females [32]. Similarly, the oviposition rate of *Adalia bipunctata* (L.) (Coleoptera: Chrysomelidae) females pre-exposed to their oviposition-deterring pheromone was similar to that of the control individuals [33]. Conversely to these precedents, our work indicates that a pre-exposure of *B. oleae* virgin females to the male-specific compound *γ*-hexalactone leads to a dose-dependent reduction in the number of eggs laid on an artificial substrate under laboratory conditions. As previously stated, OAI values lower than −0.3 indicate that testing substance is reducing the oviposition of the species, and according to our results, the number of laid eggs is reduced when females are pre-exposed to 1 mg of compound (OAI = −0.56), whereas the OAI value obtained with the lowest amount of *γ*-hexalactone was close to this threshold (OAI = −0.24). This significant reduction in the oviposition rate is observed after a 24-h pre-exposure to the compound and a subsequent egg laying period of 24 h. As reported by other authors, the prevalence of the effect mediated by a previous experience to a semiochemical is variable. For instance, the onset of the calling behavior of *Grapholita molesta* (Busck) and *Choristoneura rosaceana* (Harris) (Lepidoptera: Tortricidae) in females exposed to their sex pheromone during 24 h started earlier than in naïve females, but these changes were not observed five days later [27]. In *Spodoptera littoralis* (Boisduval) (Lepidoptera: Noctuidae), a short exposure of males to either female gland extracts or the main sex pheromone component increased their sensitivity to the sex pheromone in further olfactometer and wind tunnel trials, and this increase in the response lasted 27 h [17]. In contrast to these results, a exposure of *Bactrocera zonata* Saunders (Diptera: Tephritidae) to methyl eugenol resulted in a desensitization for the compound that lasted for at least four weeks [34]. Hence, whether the effect on *γ*-hexalactone on the oviposition of the olive fruit fly females has a short- or long-term prevalence should be further explored.

The role of *γ*-hexalactone within the chemical ecology of *B. oleae* remains still unknown, despite its proven pheromonal activity reported by López et al. [13]. Other male-specific lactones of the tephritid genera *Rhagoletis* Loew and *Anastrepha* (Schiner) have also been described as pheromones [35,36,37], although no study has aimed to determine the response of each species after a long-term exposure to these pheromonal compounds. In *B. oleae*, *γ*-hexalactone is not only attractive for virgin males and females [13], but also elicits a decrease in the oviposition rate when virgin females are exposed to the compound prior to being paired with males. It is unclear whether this decrease in the number of laid eggs may be due to either a change in the oviposition behavior of females or a disruption of mating behavior. In this regard, mating performance of the olive fruit fly can be modulated by chemical stimuli. Specifically, the mating success of both *B. oleae* males and females is increased when sexually mature individuals are exposed to *α*-pinene [38], a plant volatile that is also a component of the female-released sex pheromone in *B. oleae* [39]. Likewise, mating and egg production are enhanced after a continuous exposure to a mixture of olive volatiles [40,41], albeit the number of eggs is reduced when some constituents of this mixture (i.e., limonene and nonanal) are singly presented [40]. In light of our results, a long-term pre-exposure to *γ*-hexalactone seems to have an opposite effect to that of plant volatiles, significantly altering the oviposition behavior of the species. As the role of *γ*-hexalactone within the chemical communication of the species is not fully understood, underlying mechanisms beneath this oviposition disruption after a previous exposure remain unknown.

The seeking of eco-friendly alternatives for disrupting the oviposition of the olive fruit fly has become a matter of study, and research is available regarding the efficacy of fungicides, plant bio-stimulants and zeolite to interrupt oviposition and therefore prevent olive fruit fly infestations [42,43,44,45]. Hence, the use of semiochemicals as oviposition disrupting tools may follow the basic principle of mating disruption, in which males are unable to locate calling females when large amounts of sex pheromone are released to the environment [46,47]. So far, reports related to the use of mating disruption as a potential approach for controlling in *B. oleae* populations are scarce [48,49,50]. After a failing pilot study conducted in Spain in 1981 [48], mating disruption of *B. oleae* males was successfully achieved in another field study by releasing 5 g of olean per hectare, with a significant reduction in the infestation level [49]. In this sense, it may be tempting to speculate that a high concentration of *γ*-hexalactone in the environment may interfere in the mating and/or oviposition behavior of the species. In light of our results, some evidence of an effect on the oviposition after a previous experience to *γ*-hexalactone is suggested. However, it is worth noting that our results are supported by a limited number of samples, with only six replicates per each γ-hexalactone release rate, and therefore they should be considered as preliminary. Future research is required to gain solid knowledge about how the habituation to *γ*-hexalactone affects the behavior of the species, and specifically its oviposition. 

## Figures and Tables

**Figure 1 insects-16-00147-f001:**
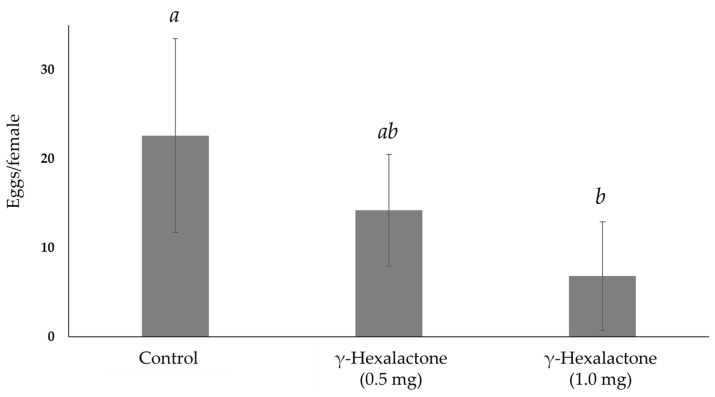
Mean number of eggs per female (±SD) of *B. oleae* pre-exposed to two amounts of *γ*-hexalactone (0.5 and 1.0 mg, n = 6 replicates per category) in comparison to non-exposed (control, n = 12) females. Columns with different letters are statistically different (Kruskal-Wallis test followed by Mann-Whitney test).

**Figure 2 insects-16-00147-f002:**
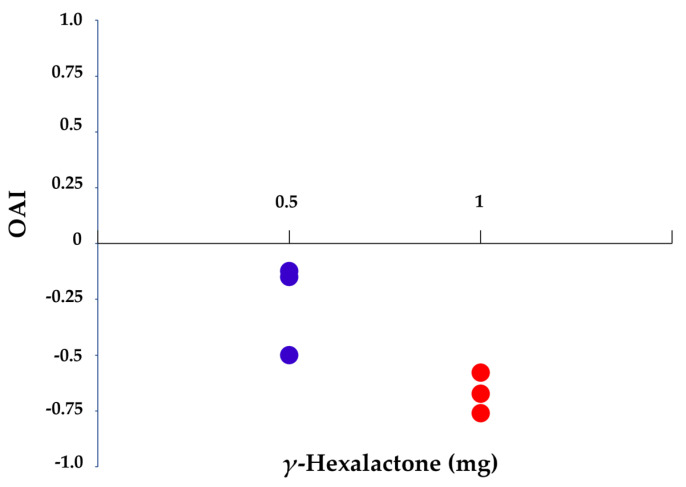
Scatter dot plot of the oviposition activity indexes (OAI) from *B. oleae* females pre-exposed to *γ*-hexalactone at 0.5 and 1.0 mg during 24 h (n = 6 per category).

## Data Availability

Data will be made available upon reasonable request.

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
