# Peer review of "A Pre-Exposure to Male-Specific Compound *γ*-Hexalactone Reduces Oviposition in *Bactrocera oleae* (Rossi) (Diptera: Tephritidae) Under Laboratory Conditions"

_insects, 2025, doi:10.3390/insects16020147_

Round 1

Reviewer 1 Report

Comments and Suggestions for Authors Title is ok
Abstract is fine
Introduction write well
in the M&M section lack of experiment
There is only 1 experiment in M&M (Oviposition assays)
In the M&M can add behavioral assay like olfactometer
If possible, to do headspace collections, field tests etc
Line 53: De Marzo et alii [14], Correct the citation.

In the discussion section required reference with Bactrocera this article may be helpful
https://link.springer.com/article/10.1007/s10340-023-01659-9

Comments on the Quality of English Language

Quality of English writing is good. 

Author Response

Title is ok

Abstract is fine

Introduction write well

in the M&M section lack of experiment

There is only 1 experiment in M&M (Oviposition assays)

In the M&M can add behavioral assay like olfactometer

If possible, to do headspace collections, field tests etc

First of all, we thank the Reviewer´s suggestions to improve the quality of our work. We agree that the manuscripts lack some additional experiments, as our results are only based on a single experiment with two gamma-hexalactone release rates. For this reason we think that such limited methodology and results herein may fit as a Short Communication. In addition, we also agree that further methodologies (i.e. headspace sampling, and behavioral and field assays) would provide a more accurate approach to understand the role of gamma-hexalactone within the chemical ecology of the species. Indeed, we reported in a previous work (López et al., 2024, J Pest Sci 97, 965–978) that gamma-hexalactone is attractive for both sexes of B. oleae in double-choice assays, and it also enhances the attractiveness of ammonium bicarbonate in field conditions. However, the main objective of this small-scale study is to particularly test whether a previous experience of females to gamma-hexalactone may affect their oviposition behavior, and therefore the experimental methodology was specifically designed to answer this question. Indeed, we would like to remark that this novel methodology for testing the oviposition behavior of B. oleae under laboratory conditions represents an advantageous alternative in absence of the natural host of the insect, that is, the olive fruit. Altogether, we feel that our results provide a preliminary basis for leading to future in-depth research aimed to determine the potential effect of a previous experience to gamma-hexalactone on different aspects of the biology of the species, such as changes in the volatile profile or behavioral responses related to intraspecific or insect-host communication, as the Reviewer suggests. In this sense, the preliminary status of our work and limitations in terms of number of samples have been highlighted in the Discussion.

Line 53: De Marzo et alii [14], Correct the citation.

We have corrected the citation and rewritten the sentence to make it more readable

In the discussion section required reference with Bactrocera this article may be helpful

https://link.springer.com/article/10.1007/s10340-023-01659-9

Following the Reviewer´s suggestion, we have introduced this reference in Lines 212 and 216 to refer to the pheromonal activity of gamma-hexalactone on B. oleae

Reviewer 2 Report

Comments and Suggestions for Authors

This study provides a small-scale experimental finding demonstrating that the specific compound γ-hexalactone, at certain concentrations, can significantly reduce the oviposition rates of Bactrocera oleae females. It is an intriguing research topic that warrants further exploration.

The number of test insects and replicates in this study appears to be relatively low. With four Bactrocera oleae females per sample and five replicates per treatment, only 20 females were tested per treatment. Comparatively, the number of test insects is clearly insufficient, which is somewhat regrettable.

Additionally, the statistical results show that the standard deviation (SD) is noticeably large, indicating substantial variation in oviposition rates across different replicates. This phenomenon suggests the need for additional replicates to better capture the pattern of oviposition behavior. A larger number of observations often results in a normal distribution, allowing for a more accurate understanding of stable trends.

If possible, the discussion should be supplemented with further elaboration on these issues.

Please double-check if there are any errors in "De Marzo et alii" on line 53.

Author Response

This study provides a small-scale experimental finding demonstrating that the specific compound γ-hexalactone, at certain concentrations, can significantly reduce the oviposition rates of Bactrocera oleae females. It is an intriguing research topic that warrants further exploration.

Comment 1: The number of test insects and replicates in this study appears to be relatively low. With four Bactrocera oleae females per sample and five replicates per treatment, only 20 females were tested per treatment. Comparatively, the number of test insects is clearly insufficient, which is somewhat regrettable.

Additionally, the statistical results show that the standard deviation (SD) is noticeably large, indicating substantial variation in oviposition rates across different replicates. This phenomenon suggests the need for additional replicates to better capture the pattern of oviposition behavior. A larger number of observations often results in a normal distribution, allowing for a more accurate understanding of stable trends.

Response 1: We thank the Reviewer for his/her comments to improve our work. We totally agree that our work is a small-scale study, limited to only six replicates per each gamma-hexalactone release rate. Indeed, there is an error related to the number of samples. Regarding gamma-hexalactone samples, 6 replicates were tested (as stated in line 169) instead of five (line 130), while 12 control samples were run (one control sample per each replicate) instead of only six.  This information has been updated in Lines 130-132 and 162-163.  

It is also true that testing more replicates would have allowed us to draw more robust conclusions about the disruptant effect on oviposition after a previous experience to the compound. However, some previous works on tephritid species have showed that testing a similar number of insects to our may lead to conclusive results in oviposition assays. For instance, Kovaiou and coworkers (2024, Insects 15: 256) evaluated the effect of high-quality zeolite as an oviposition-disruptant by mean of different experiments considering five females per replicate, for a total of four replicates, in each experiment. Similarly, the effect of the secondary plant metabolite phloroglucinol on the oviposition rate Zeugodacus cucurbitae was evaluated by testing six replications of each three males and three females (Puri et al., 2022, Comp. Biochem. Physiol. C, 109291). In our case, and in spite of the low number of sampling females and the standard deviation, a 24-h pre-exposure of virgin females to gamma-hexalactone appears to exert a reduction in the number of eggs laid, and this effect is partially supported by the statistical analysis and the oviposition activity index. Nonetheless, we agree that additional replicates would have provided a more solid basis for our conclusions, and this should be taken into account for further studies aimed to delve into the effect on the behavior of the species after a previous experience to the compound.

Comment 2: If possible, the discussion should be supplemented with further elaboration on these issues.

Response 2: Following the Reviewer´s suggestion, previous issues have been addressed in the discussion, to remark that our results should be considered as preliminary basis and limitations of our results.

Comment 3: Please double-check if there are any errors in "De Marzo et alii" on line 53.

Response 3: We have corrected the citation and rewritten the sentence to make it more readable

Round 2

Reviewer 1 Report

Comments and Suggestions for Authors

Manuscript Feedback:

 - Structure: Ensured logical flow with a clear introduction, methods, results, and discussion.

- References: Were accurate and relevant, including recent and seminal works.

- Visuals: Were clear, well-labeled, and referenced in the text.

- Language: Was clear, concise, and jargon-free unless necessary.

- Methods: Addressed flaws and justified design choices.

- Novelty: Significance and practical implications.

- Reviews: Addressed all comments in the revised manuscript.

- Ethics: Confirmed compliance without issue.

- Proofreading: Eliminated errors; formatting consistency was ensured by the journal.

Comments on the Quality of English Language

Manuscript Feedback:

- Structure: Ensured logical flow with a clear introduction, methods, results, and discussion.

- References: Were accurate and relevant, including recent and seminal works.

- Visuals: Were clear, well-labeled, and referenced in the text.

- Language: Was clear, concise, and jargon-free unless necessary.

- Methods: Addressed flaws and justified design choices.

- Novelty: Significance and practical implications.

- Reviews: Addressed all comments in the revised manuscript.

- Ethics: Confirmed compliance without issue.

- Proofreading: Eliminated errors; formatting consistency was ensured by the journal.